# Identification and Characterization of Petal Color Change from Pink to Yellow in *Chrysanthemum morifolium* 'Pink Candy' and Its Bud Variant

**Lian-Da Du** [1], **Yan-Hong Liu** [1], **Jin-Zhi Liu** [1], **Xiang-Qin Ding** [1], **Bo Hong** [2], **Da-Gang Hu** [1] and **Cui-Hui Sun** [1,*]

[1]  National Key Laboratory of Crop Biology, College of Horticulture Science and Engineering, Shandong Agricultural University, Tai'an 271018, China
[2]  Beijing Key Laboratory of Development and Quality Control of Ornamental Crops, Department of Ornamental Horticulture, China Agricultural University, Beijing 100193, China
*  Correspondence: suncuihui@sdau.edu.cn

**Abstract:** Chrysanthemum, one of the most popular ornamental plants in the world, is renowned for its brilliant colors and multifarious flower types. Thousands of gorgeous chrysanthemum cultivars exist thanks to both traditional breeding techniques and its characteristic bud sporting. In this study, we identified a pink-to-yellow flower color-changed bud sport of the edible chrysanthemum cultivar 'Pink Candy'. The bud variant and its parent plant bloomed at the same time, but with yellow- and pink-colored flowers, respectively. However, the two flower types exhibited strikingly different combinations and concentrations of primary and secondary metabolites, aromatic compounds, and pigments. Additionally, the expression patterns of key pigment biosynthesis genes, such as *CmPAL* (*phenylalanine ammonia lyase*), *CmDFR* (*dihydroflavonol 4-reductase*), *CmF3H* (*flavanone 3'-hydroxylase*), *CmNXS* (*neoxanthin synthase*) and *CmCCD4* (*carotenoid cleavage dioxygenase 4*) were distinct between both flower types, helping to explain the color transformation of the mutant to some extent. Taken together, our results suggest a mechanism explaining the transformation of pink flowers to yellow flowers in the mutant bud sport. These results provide the foundation for the production of a novel chrysanthemum cultivar.

**Keywords:** bud sport; flower color; flavor compound; pigment biosynthesis; gene expression; chrysanthemum

## 1. Introduction

The flower, or inflorescence, is the most valuable and attractive part of ornamental plants. Ornamental qualities comprise petal color, petal shape and size, flower fragrance, shelf life, and so on; however, flower color is of particular interest in both the biologically essential merits as well as the commercially determinant traits of ornamental plants [1]. In general, flower color is determined by intrinsic factors, such as pigments, vacuolar pH, metal ion complexation, epidermal cell shapes of the petal, and genetics [2–4], as well as extrinsic factors, such as temperature, light, and even cultivation management [5,6]. Untangling the complex mechanism responsible for flower coloration is essential for precision breeding of novel ornamental flowering plants.

Ornamental plants are of great importance for esthetic function with a super-broad color spectrum of flower colors [7]. Although there are so many colorful flowers in nature, some colors are confined to certain species of ornamental plants. Thus, the improvement and introduction of novel flower colors have been a central goal for flower breeders [8]. For a long period, much progress has been made in ornamental flower color breeding. Flower breeders have successfully created a plethora of novel ornamental varieties using traditional breeding technologies, including the use of complex species crosses, which have been conducive to creating plenty of new varieties of ornamental species for decades [9].

The development of mutation breeding techniques has further allowed for the efficient production of a large number of novel flower colors and combinations in ornamental plants [10]. For example, the application of physical and chemical mutagens has been used to produce novel colors in chrysanthemum and kalanchoe [11,12]. Ion beam irradiation has successfully produced novel colors in rose [13], petunia [14], and geranium [15]. Most recently, genetic engineering has been successfully employed to create a variety of novel color variants, including blue-hued roses [16], splatter-spotted lilies [17], and pale-blue torenias [18].

In addition to the above-mentioned flower breeding methods, one spontaneous mutation method, known as bud sports, which occur naturally and randomly in many vegetatively propagating plants, has been used to breed new cultivars for centuries. A bud sport refers to an occasional mutation of a lateral bud, an inflorescence, a branch, or single flower/fruit with an obviously different phenotype from the parent plant [7]. Bud sports are primarily caused by stable somatic mutations and can thus be passed through vegetative propagation to clonal descendants [19]. For a long period, many popular and commercially successful agricultural plant varieties have been produced from spontaneously generated bud sports, including certain varieties of grape, apple, peach, and pummelo [20–23]. Similarly, sports have generated numerous novel ornamental plant varieties, including the hooked-petal chrysanthemum 'Anastasia Dark Green' [24], the ornamental peach 'Hongbaihuatao' with pink, red, and variegated flowers on one branch [25], as well as various color sports of azalea (*Rhododendron simsii* hybrids) [26].

Chrysanthemum (*Chrysanthemum morifolium*), a perennial herb with inflorescence comprised of marginal ray florets and central disc florets [24], is renowned as one of the wonders of the flower world due to the staggering diversity exhibited by its flowers. It is believed that the heterozygous genetic background present in chrysanthemums is the result of innumerable crossings and interspecific hybridization events. Because of this variability, most stable chrysanthemum varieties are propagated vegetatively. Specifically, bud sports have been utilized to produce both novel and stable chrysanthemum varieties for over 1600 years [27,28]. Here, we discovered a flower color altered bud sport of one chrysanthemum cultivar ('Pink Candy', short for PC) occurring occasionally during its vegetative propagation and cultivation. The bud sport flowered at the same time as its parent plant PC, but with bright yellow-colored ray florets, different from the pink-colored ray florets of PC. We further sought to investigate the molecular mechanism responsible for the altered flower phenotype, by assessing changes in primary and secondary metabolites, aroma and pigment compounds, and expression patterns of pigment biosynthesis genes.

## 2. Materials and Methods

### 2.1. Plant Materials

The pink-colored chrysanthemums (*C. morifolium* 'Pink Candy') were obtained from China Agricultural University (Beijing, China) and were grown in the chrysanthemum nursery of the Shandong Agricultural University (Shandong, China). A branch of yellow flower colored mutant (YM) chrysanthemum was found on the pink flower colored 'Pink Candy' (PC). After reproduction, the two types of materials were grown under the same conditions. The ray floret samples of both chrysanthemums were collected at the full-bloom stage for further analysis.

### 2.2. Measurement of Primary Substances in the Ray Florets

The content of floral amino acids was determined according to the method of Li et al. [29]. Approximately 0.3 g of fresh ray florets was ground with 3 mL of 10% acetic acid, and distilled water was added to make a volume of 25 mL. Then, 1 mL of the supernatant was mixed with 0.1 mL of 0.1% ascorbic acid solution and 3 mL of ninhydrin, and the solution was heated for 30 min in an 80 °C water bath. Finally, enough 60% ethanol was added to bring the total volume to 20 mL, and 1 mL of the final mixture was analyzed by absorbance spectrophotometry ($OD_{570\,nm}$).

Soluble protein content was determined using the Coomassie Brilliant Blue G-250 staining method described by Eris et al. [30]. About 0.3 g of fresh ray florets was ground with 5 mL of distilled water. The mixture was centrifuged at 3000 r/min for 10 min, and the supernatant was collected and diluted in a 20 mL volumetric flask. Finally, a 0.5 mL subsample of the diluted solution was mixed with 5 mL of Coomassie Brilliant Blue G-250 dye. The final mixture was analyzed by absorbance spectrophotometry ($OD_{595\,nm}$).

To measure the content of floral soluble sugars, 0.2 g of fresh ray florets was ground and extracted twice in an 85 °C water bath for 30 min. Next, 0.1 mL of the supernatant was mixed with 0.3 mL of distilled water, 0.1 mL of anthranone (20 mg/mL), and 1 mL of strong sulfuric acid, and the mixture was heated at 100 °C for 1 min. After cooling, 1 mL of the solution was analyzed by absorbance spectrophotometry ($OD_{630\,nm}$).

The total phenolic content was estimated using the Folin–Ciocalteu reagent as described by Katarzyna et al. [31]. About 1 g of ground floral powder was mixed with 25 mL of 60% ethanol and 0.2% HCl. The mixture was subsequently shaken vigorously at 100 Hz at 45 °C in dark for 60 min. Subsequently, the solution was analyzed by absorbance spectrophotometry ($OD_{722\,nm}$). The total polyphenol content was calculated using the calibration curve created using gallic acid as the reference standard.

All the assays were performed in triplicate.

### 2.3. Analysis of Floral Aroma Compounds

Floral aroma compounds were characterized using gas chromatography–mass spectrometry (GC-MS) [32]. Briefly, ray florets were weighed and placed in an individual, sealed conical flask RTX-1MS (30 m × 0.25 mm × 0.25 mm) column with helium (99.999%) as the carrier gas. The following parameters were used: 1 µL injection volume, 2.18 mL·min$^{-1}$ flow rate, 10:1 split ratio, 800 eV electron impact (EI) energy, 230 °C ion source temperature, and 150 °C inlet temperature. EI mass spectra were recorded in the 45–450 amu range at 1 s intervals. Quantitative analysis was performed using 10 µL of 69.32 ng·µL$^{-1}$ ethyl decanoate as the internal standard. All assays were performed in triplicate.

### 2.4. Chromogenic Reaction of Floral Pigment Compounds

Floral pigment compounds were determined using extraction and chromogenic reaction. Ray florets were ground in liquid nitrogen to a fine powder, and the powder was subsequently subjected to two separate extractions. First, 0.1 g of fine powder was placed in a dry test tube to which 5 mL each of petroleum ether, 10.0% hydrochloric acid (HCl), and 30.0% $NH_3 \cdot H_2O$ were added. The solution was thoroughly mixed and filtered through a 0.22 µm organic membrane. Second, 0.1 g of fine powder was placed in a dry test tube with 10 mL of saline-acidified methanol (HCl:methanol = 1:99, *v/v*) and extracted for 15 h. After filtration, saline-acidified methanol was added to make a total volume of 25 mL. Filtered solutions were used for observation of the chromogenic reactions.

Aluminum trichloride chromogenic reaction: 2 mL of extracted solution was mixed with 1 mL of 1% $AlCl_3 \cdot 6H_2O$ methanol solution and used for color observation. Additionally, a sulfuric acid chromogenic reaction was performed: 2 mL of extracted solution was mixed with 1.5 mL of vitriol oil and placed in boiling water for 5 min prior to color observation.

Lead acetate chromogenic reaction: 2 mL of extracted solution was mixed with 2 mL of 1% $(CH_3COO)_2Pb \cdot 3H_2O$ methanol solution and used for color observation.

Ferric trichloride chromogenic reaction: 2 mL of extracted solution was mixed with 2 mL of 5% $FeCl_3 \cdot 6H_2O$ methanol solution and used for color observation.

Sodium tetrahydroborate chromogenic reaction: 2 mL of extracted solution was mixed with 8 mg $NaBH_4$ and 2 mL of 1% HCl and reacted for 1 h prior to color observation.

### 2.5. Analysis of Floral Pigments

The content of floral total flavonoids was determined according to the method of Liu et al. [33]. Briefly, 0.2 of ray floret powder was mixed with 3 mL of methanol and extracted in the dark for 24 h. The solution was then filtered through a 0.22 µm organic membrane.



Subsequently, 0.3 mL of the filtered solution was mixed with 4.7 mL of 1% $AlCl_3 \cdot 6H_2O$-methanol solution, and allowed to stand for 10 min at room temperature. Subsequently, the solution was analyzed by absorbance spectrophotometry ($OD_{405 \text{ nm}}$) (UV-2450, Shimadzu, Kyoto, Japan). All assays were performed in triplicate.

Total floral anthocyanins were extracted using the methanol-HCl method and detected as described by Hu et al. [34]. Briefly, 0.5 g of fresh ray florets was ground and mixed with 5 mL of 1% HCl and methanol solution, and subsequently extracted in a 32 °C water bath for 4 h. After extraction, the supernatant was used for absorbance spectrophotometry ($OD_{340 \text{ nm}}$ and $OD_{657 \text{ nm}}$) (UV-2450, Shimadzu, Japan). The total anthocyanin content was calculated using a relative formula. All the assays were performed in triplicate.

The total floral carotenoid content was determined according to a previously published protocol [35]. Briefly, 0.2 g of fresh ray florets was mixed with 20 mL of 95% ethanol, and then extracted at 25 °C for 24 h. The mixture was subsequently centrifuged at 4000 r/min for 10 min. The supernatant was used for absorbance spectrophotometry ($OD_{340 \text{ nm}}$, $OD_{657 \text{ nm}}$, and $OD_{663 \text{ nm}}$) (UV-2450, Shimadzu, Japan). The total carotenoid content was calculated using a relative formula. All assays were performed in triplicate.

### 2.6. Analysis of Floral Secondary Metabolites

The content of several floral secondary metabolites, including apigenin, naringenin, galuteolin, baicalin, chlorogenic acid, and isochlorogenic acid, was determined using a modified high-performance liquid chromatography (HPLC) method. Briefly, ray florets were ground thoroughly in liquid nitrogen, and 1 g of the resultant fine power was filtered and extracted to prepare the sample. The mobile phase consisted of acetonitrile (A) and water (B), and filtration was performed using a 0.2 μm hydrophilic polypropylene filter. The separation was performed using a gradient program, as previously described [36]. Analysis was performed using a $C_{18}$ reversed phase column, with an oven temperature of 25 °C, a flow rate of 1 mL min$^{-1}$, and a 20 μL injection volume.

### 2.7. Quantitative Real-Time PCR (qRT-PCR)

Total RNA was extracted from both YM and PC ray florets using TRIzol Reagent (Vazyme, Nanjing, China). Reverse transcription (RT) was conducted using a PrimeScript first-strand cDNA synthesis kit (TaKaRa, Dalian, China). The relative expression levels of anthocyanin- and carotenoid-related genes were calculated using the $2^{-\Delta\Delta Ct}$ method [37]. Primers used for qRT-PCR are listed in Supplementary Table S1. All assays were performed in triplicate.

### 2.8. Statistical Analysis

All assays were performed in triplicate and represented as the mean ± standard deviation. Statistical significance was assessed using Duncan's multiple range test ($p < 0.05$).

## 3. Results

### 3.1. Discovery of a Novel Floral Bud Sport in Chrysanthemum 'Pink Candy'

We discovered a yellow-flowered bud sport (YM) on the pink-flowered chrysanthemum cultivar 'Pink Candy' (PC) (Figure 1A). Specifically, the pink flowers of PC plants became less saturated in hue as the flowering stage progressed, and this phenomenon was particularly evident in the outer layers of ray florets. The YM bud sport exhibited bright yellow flowers, with the outer layers of ray florets less saturated in hue (Figure 1B). Aside from flower color, YM and PC exhibited no other obvious differences (Figure 1C). As the branch of the different flower color mutant sprouted on only one chrysanthemum plant, we had reason to believe that YM was a natural bud mutation, which can easily occur on chrysanthemums during normal maintenance. This novel floral discovery prompted us to propagate the YM bud sport for further analysis.

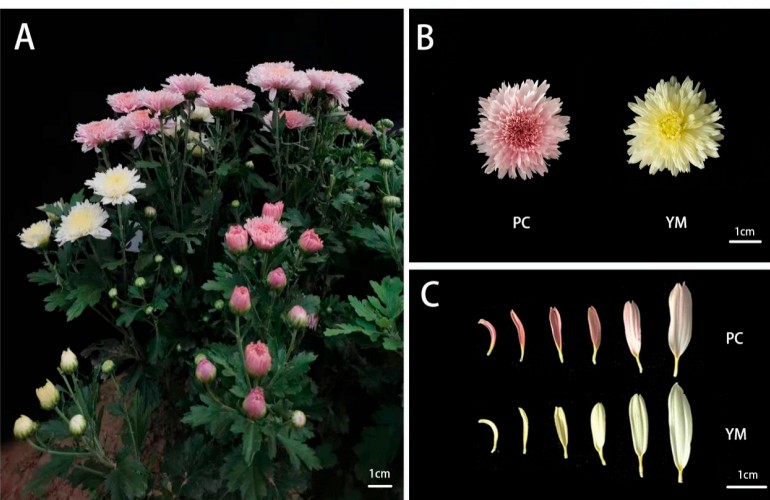

**Figure 1.** Phenotypic differences between 'Pink Candy' (PC) and yellow-colored mutant (YM) chrysanthemum flowers. (**A**) The phenotypes of PC and YM under standard cultivation conditions. (**B**) Top view of inflorescences of PC and YM at the full-bloom stage. (**C**) The phenotypes of different layers of ray florets of PC and YM. Bar = 1 cm.

*3.2. Differences in Floral Primary Substances and Secondary Metabolites between PC and YM*

As the flower color was the only obvious difference between PC and YM in outer appearance, in order to identify the underlying chemical differences, the content of primary substances in the ray florets of PC and YM was estimated. Compared to PC flowers, YM flowers contained a slightly higher content of total amino acids (Figure 2A), a considerably higher content (+35.27%) of soluble sugars (Figure 2B), a higher content of polyphenols (Figure 2C), and a higher content of soluble protein (Figure 2D). As the four types of primary substances displayed various degrees of differences between PC and YM, this suggested some functional and adaptive changes might occur between PC and YM.

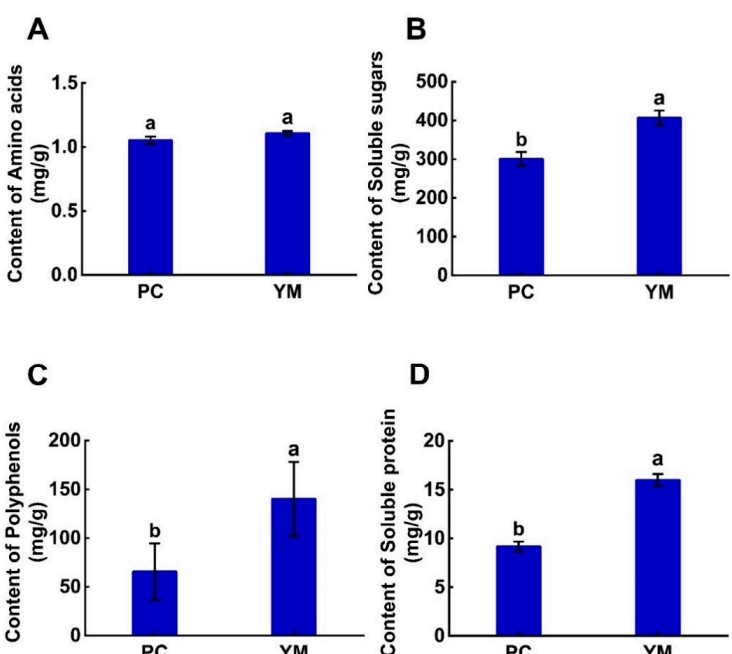

**Figure 2.** The analysis of primary metabolites in the ray florets of PC and YM. The content of amino acids (**A**), soluble sugars (**B**), polyphenols (**C**), and soluble protein (**D**) in ray florets of PC and YM. Different letters indicate significant differences at $p < 0.05$ according to Duncan's multiple range test.

Furthermore, the chrysanthemum 'Pink Candy' was selected by China Agricultural University for its edibility and medicinal value, so we sought to examine differences in several secondary metabolites between PC and YM. YM flowers contained 62.5% less apigenin than PC flowers (Figure 3A). Conversely, YM flowers contained 58.05% more naringenin, 98.73% more galuteolin, and 93.24% more baicalin than PC flowers (Figure 3B–D). Both PC and YM flowers contained nearly the same amount of chlorogenic acid, isochlorogenic acid, acacetin, quercetin, rutin, hesperetin, and luteolin (Figure 3E–K). In term of these secondary metabolites, there were few differences between PC and YM, which suggested the almost identical medicinal value of PC and YM.

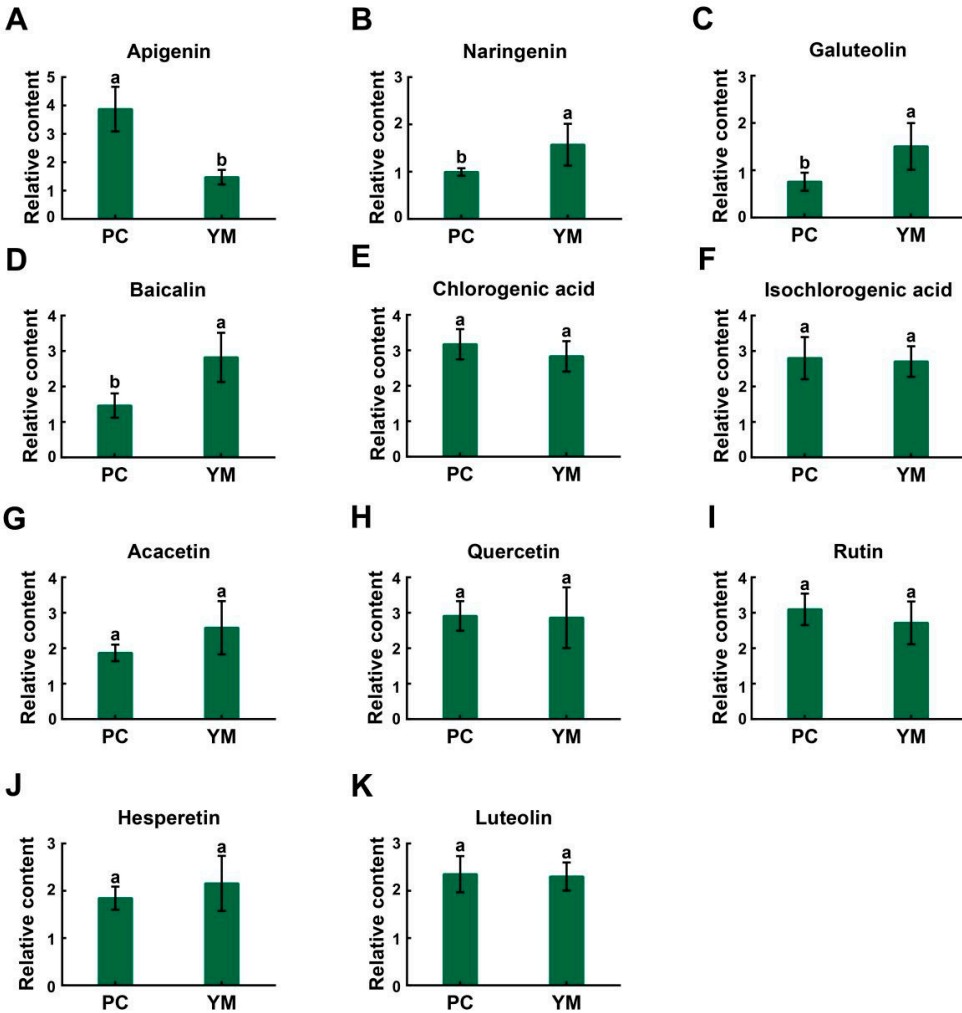

**Figure 3.** The analysis of secondary metabolites in the ray florets of PC and YM. The relative content of apigenin (**A**), naringenin (**B**), galuteolin (**C**), baicalin (**D**), chlorogenic acid (**E**), isochlorogenic acid (**F**), acacetin (**G**), quercetin (**H**), rutin (**I**), hesperetin (**J**), and luteolin (**K**) in ray florets of PC and YM. Different letters indicate significant differences at $p < 0.05$ according to Duncan's multiple range test.

*3.3. Differences in Floral Aromatic Compounds between PC and YM*

To further explore the intrinsic differences between PC and YM, aroma determination of inflorescence of PC and YM was conducted. We divided floral aroma compounds into four broad categories: terpenes, alcohols, ketones, and aldehydes. Both PC and YM flowers contained nearly identical numbers of compounds in each category (Figure 4A). Additionally, both PC and YM flowers contained nearly identical content of terpenes and aldehydes. However, PC flowers contained a significantly higher content of alcohols and ketones than YM flowers (Figure 4B). Notably, although both flower types contained almost identical numbers of aromatic compounds, the identities of those compounds were quite

different (Figure 4C, Table 1). In PC flowers, the aromatic compounds with the three highest concentrations were 2-hexenal (E), eucalyptol, and bicyclo [3.1.0]hex-3-en-2-one,4-methyl-1(1-methylethyl), while (E)-beta-famesene had the lowest concentration. Interestingly, YM flowers contained two unique compounds: cyclohexene,3-(1,5-dimethyl-4-hexenyl)-6-methylene-,[S-(R*, S*)-] and bicyclo [3.1.0] hexane,4-m ethylene-1-(1-methylenethyl). Aroma determination assays showed that the concentrations of specific flavor substances in inflorescences of PC and YM varied greatly, which suggested that PC and YM flowers were likely to have different fragrances.

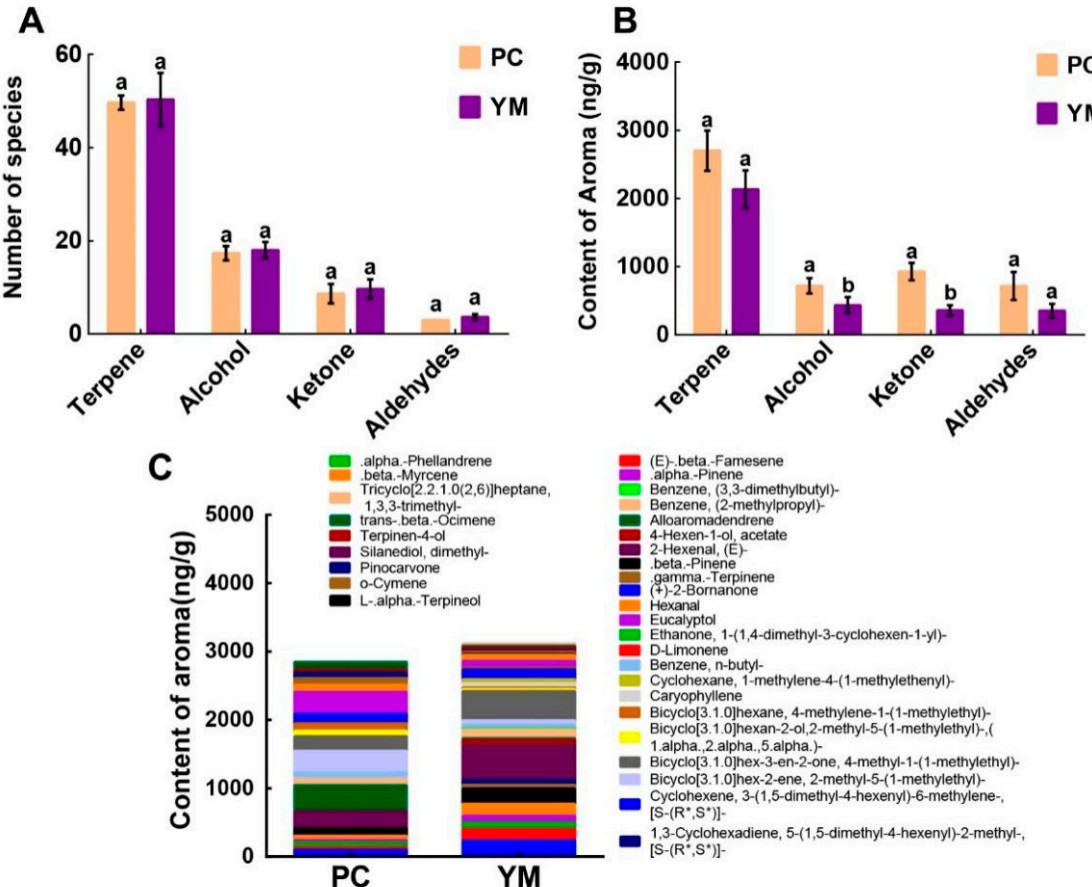

**Figure 4.** The analysis of aroma compounds in the ray florets of PC and YM at the full-bloom stage. (**A**) The number of species of four types of aroma compounds in ray florets of PC and YM. (**B**) The content of species of four types of aroma compounds in ray florets of PC and YM. (**C**) The content of the specific aroma components in ray florets of PC and YM.

**Table 1.** The content of flavor compounds in fresh flowers of PC and YM.

| Compound Name | Molecular Formula | Retention Time | Aroma Content (ng/g) | |
|---|---|---|---|---|
| | | | PC | YM |
| (+)-2-Bornanone | $C_{10}H_{16}$ | 13.20 | 100.53 | 237.47 |
| (E)-.beta.-Famesene | $C_{10}H_{14}$ | 20.31 | 36.22 | 167.36 |
| .alpha.-Phellandrene | $C_6H_{12}O$ | 9.54 | 66.19 | 90.77 |
| .alpha.-Pinene | $C_{10}H_{18}O$ | 7.67 | 36.86 | 99.06 |
| .beta.-Myrcene | $C_{15}H_{24}$ | 9.20 | 63.19 | 175.87 |
| .beta.-Pinene | $C_{10}H_{16}$ | 8.80 | 104.54 | 222.42 |
| .gamma.-Terpinene | $C_{10}H_{16}$ | 10.98 | - | 54.93 |
| 1,3-Cyclohexadiene, 5-(1,5-dimethyl-4-hexenyl)-2-methyl-, [S-(R*,S*)]- | $C_{15}H_{24}$ | 21.15 | - | 72.73 |
| 2-Hexenal, (E)- | $C_{10}H_{18}O$ | 5.66 | 253.78 | 505.98 |
| 4-Hexen-1-ol, acetate | $C_8H_{14}O_2$ | 5.71 | - | 78.48 |
| Alloaromadendrene | $C_{10}H_{14}O$ | 20.99 | 383.10 | 26.79 |
| Benzene, (2-methylpropyl)- | $C_{10}H_{16}$ | 7.97 | 93.21 | 128.11 |
| Benzene, (3,3-dimethylbutyl)- | $C_{12}H_{18}S$ | 6.03 | - | 15.12 |
| Benzene, n-butyl- | $C_{10}H_{14}$ | 7.96 | 91.17 | 51.81 |
| Bicyclo [3.1.0]hex-2-ene, 2-methyl-5-(1-methylethyl)- | $C_{15}H_{24}$ | 7.50 | 317.11 | 64.66 |
| Bicyclo [3.1.0]hex-3-en-2-one, 4-methyl-1-(1-methylethyl)- | $C_6H_{10}O$ | 13.92 | 211.09 | 428.54 |
| Bicyclo [3.1.0]hexan-2-ol, 2-methyl-5-(1-methylethyl)-, (1.alpha.,2.alpha.,5.alpha.)- | $C_{10}H_{16}$ | 11.20 | 80.36 | 27.86 |
| Bicyclo [3.1.0]hexane, 4-methylene-1-(1-methylethyl)- | $C_{10}H_{16}$ | 8.73 | 108.88 | 21.94 |
| Caryophyllene | $C_{15}H24$ | 19.63 | - | 63.17 |
| Cyclohexane, 1-methylene-4-(1-methylethenyl)- | $C_{10}H_{16}$ | 10.19 | - | 58.04 |
| Cyclohexene, 3-(1,5-dimethyl-4-hexenyl)-6-methylene-, [S-(R*,S*)]- | $C_{10}H_{16}$ | 21.74 | 139.14 | 139.14 |
| D-Limonene | $C_{10}H_{16}$ | 10.19 | - | 15.82 |
| Ethanone, 1-(1,4-dimethyl-3-cyclohexen-1-yl)- | $C_{10}H_{16}O$ | 10.12 | - | 12.54 |
| Eucalyptol | $C_{10}H_{16}$ | 10.25 | 322.42 | 101.36 |
| Hexanal | $C_{15}H_{24}$ | 4.42 | 102.37 | 91.29 |
| L-.alpha.-Terpineol | $C_{10}H_{18}O$ | 14.36 | - | 11.67 |
| o-Cymene | $C_{10}H_{17}$ | 10.08 | 100.10 | 33.95 |
| Pinocarvone | $C_{10}H_{14}O$ | 13.66 | 82.23 | 19.29 |
| Silanediol, dimethyl- | $C_2H_8O_2Si$ | 2.60 | - | 12.12 |
| Terpinen-4-ol | $C_{10}H_{18}O$ | 14.03 | 45.03 | 35.20 |
| trans-.beta.-Ocimene | $C_{10}H_{16}$ | 10.43 | 111.95 | 20.87 |
| Tricyclo [2.2.1.0(2,6)]heptane, 1,3,3-trimethyl- | $C_{10}H_{16}$ | 10.19 | - | 32.61 |

Note: '-' means not tested.

### 3.4. Differences between Floral Pigment Compounds in PC and YM

As the most obvious difference between PC and YM flowers was their flower color, we sought to probe the cause of color difference between the two materials. To that end, several chromogenic reaction assays were performed. The petroleum ether reaction for both PC and YM flowers resulted in a faint yellow color, indicating the presence of carotenoids (Supplementary Figure S1A). The HCl reaction resulted in a light pink color for PC flowers, indicating the presence of anthocyanins (Supplementary Figure S1B). The $NH_3 \cdot H_2O$ reaction resulted in a light brown color for PC flowers and a yellow color for YM flowers, indicating that PC flowers contained a higher flavonoid content than YM flowers (Supplementary Figure S1C). Both the $AlCl_3$ and sulfuric acid reactions resulted in a yellow color for both PC and YM flowers, although of different intensities, indicating the presence of flavonoids (Supplementary Figure S1D,G). The lead acetate reaction resulted in a turquoise precipitate for PC flowers, indicating the presence of anthocyanins, and a white precipitate for YM flowers, indicating the presence of flavonoids with either o-diphenol hydroxyl groups, 4-keto groups, or 5-hydroxy groups (Supplementary Figure S1E). The $FeCl_3$ reaction indicated that neither PC nor YM contained any pigments with phenolic hydroxyl groups (Supplementary Figure S1F). The $Na_2B_4O_7$ reaction indicated that neither PC nor YM contained either flavonones or flavanonols (Supplementary Figure S1F). The $Na_2CO_3$ reaction resulted in a light yellow color for both PC and YM flowers, indicating the presence of flavonoids, but not specifically dihydroflavone or chalcone (Supplementary Figure S1I). Both the $H_3BO_3$ and $SrCl_2$ reactions indicated that neither PC nor YM contained either flavonoids with $C_5$-OH groups or $3',4'$-dihydroxy pigments (Supplementary Figure S1J,K).

The chromogenic reaction assays showed there were some of differences between the two flower types regarding pigment types and concentrations, which may be the potential cause of the color difference between PC and YM. To confirm the presence of flavonoids, anthocyanins, carotenoids, and lutein in PC and YM flowers, we quantitatively assessed their respective concentrations. PC flowers contained approximately six times more and four times more total anthocyanins than YM flowers (Figure 5A,B). However, YM flowers contained approximately 150.89% more total carotenoids and 61.45% more lutein than PC flowers (Figure 5C,D). Clearly, and logically, the pink coloration of PC flowers stems from higher proportions of flavonoids and anthocyanins, while the yellow coloration of YM flowers stems from higher proportions of carotenoids and lutein.

### 3.5. Differences in Floral Gene Expression between PC and YM

Petal pigmentation is closely related to the transcript levels of key structural genes linked to the biosynthesis of flavonoids and carotenoids, including *CHS*, *CHI*, *F3H*, *F3'H*, *DFR*, *ANS*, *PDS*, *PSY*, *LCYE*, *ECH*, and *CCD* [38,39]. To understand the molecular mechanism underlying flower color in PC and YM, we performed qRT-PCR analysis. Compared to PC flowers, the first key gene in flavonoid biosynthesis, *CmPAL*, was significantly upregulated in YM flowers. However, several genes linked to procyanidin biosynthesis, *CmF3H*, *CmFLS*, and *CmDFR*, were significantly downregulated in YM flowers (Figure 6A), supporting our previous floral pigment assays that indicated a lack of anthocyanins in YM flowers. Both PC and YM flowers exhibited patterns distinctive of carotenoid biosynthesis gene expression. Both *CmLCYE* and *CmECH* were found to be significantly upregulated in YM flowers, supporting our previous result indicating that YM flowers contain a greater proportion of lutein than PC flowers. Meanwhile, *CmPSY*, *CmPDS*, *CmNXS*, and *CmCCD4* were significantly downregulated in YM flowers, again supporting our previous result indicating that YM flowers contain a greater proportion of carotenoids than PC flowers.

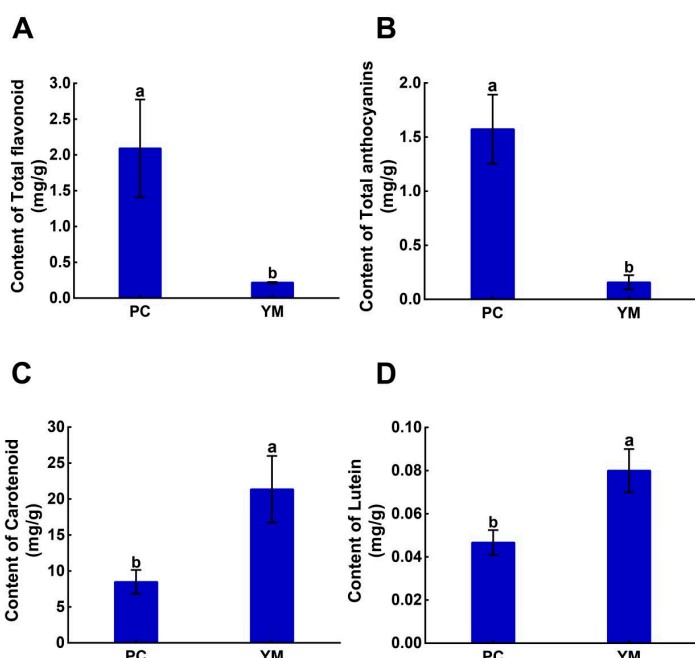

**Figure 5.** The content of pigments in the ray florets of PC and YM. The content of total flavonoids (**A**), total anthocyanins (**B**), carotenoids (**C**), and lutein (**D**) in ray florets of PC and YM. Different letters indicate significant differences at $p < 0.05$ according to Duncan's multiple range test.

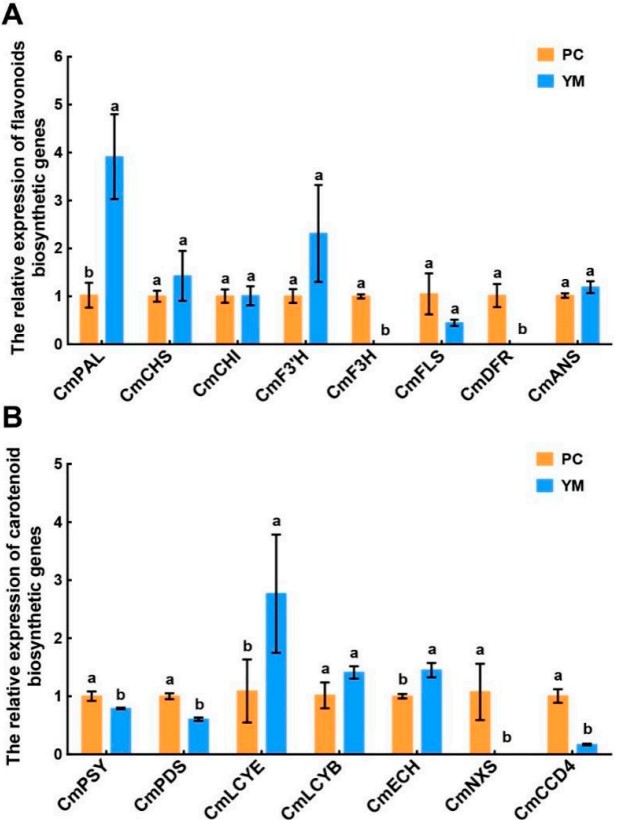

**Figure 6.** The relative expression levels of genes related to the biosynthesis of flavonoids (**A**) and carotenoids (**B**) in the ray florets of PC and YM. Different letters indicate significant differences at $p < 0.05$ according to Duncan's multiple range test.

### 3.6. Proposed Mechanism of Pink-to-Yellow Flower Color Transformation in Chrysanthemum 'Pink Candy'

In this study, we discovered one bud sport YM with yellow-colored flowers, which was different from the pink flower colored parent plant PC. This study focused on the identification of differences of ray florets between PC and YM. Based on the correlation analysis of detected metabolites and the expression levels of related genes in PC and YM, we concluded that the secondary metabolites baicalin, galuteolin, and naringenin shared a high positive correlation, and that acacetin and galuteolin were also positively correlated with each other. The content of flavonoids was positively correlated with the content of both alcohols and terpenes. The expression level of *CmF3H* was positively correlated with aldehyde content and the expression level of *CmPAL* was positively correlated with lutein content; however, the expression level of *CmLCYE* was negatively correlated with the content of flavonoids, alcohols, terpenes, and lutein. The expression levels of *CmPDS* and *CmANS* were negatively correlated with the content of amino acids and aldehydes, respectively. Finally, the expression level of *CmNXS* was negatively correlated with that of *CmLCYB*, *CmF3H*, and *CmANS* (Figure 7A).

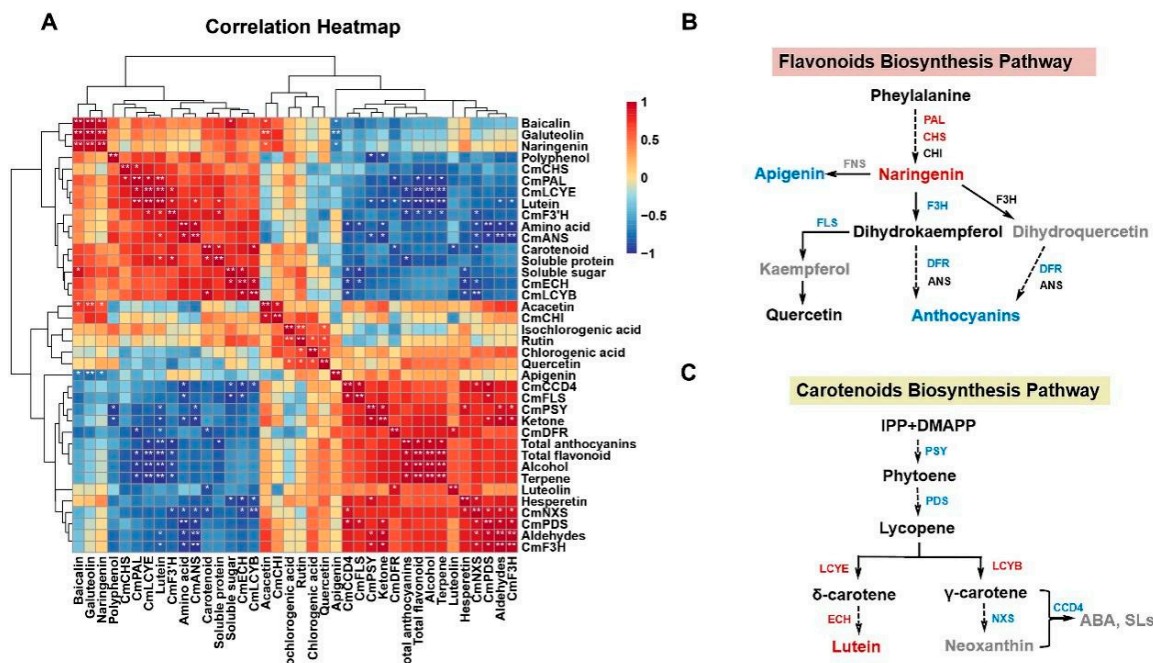

**Figure 7.** (**A**) The correlation analysis of detected metabolites and expression levels of related genes in PC and YM. (**B**,**C**) The working model of flower color transformation of YM. Red coloration means increased metabolites and gene expression in YM flowers compared to PC flowers, while blue coloration means decreased metabolites and gene expression in YM flowers. Gray coloration indicates the undetected metabolites in this study. Abbreviations: PAL, phenylalanine ammonia lyase; CHS, chalcone synthase; CHI, chalcone isomerase; FNS, flavone synthase; F3H, flavanone 3′-hydroxylase; FLS, flavonol synthase; DFR, dihydroflavonol 4-reductase; ANS, anthocyanidin synthase; PSY, phytoene synthase; PSD phytoene desaturase; LCYE, lycopene ε-cyclase; LCYB, lycopene β-cyclase; ECH, ε-carotenoid hydroxylase; NXS, neoxanthin synthase; CCD4, carotenoid cleavage dioxygenase 4.

Taken together, these results suggested a mechanism for pink-to-yellow flower color transformation in chrysanthemum 'Pink Candy'. In YM flowers, genes linked to the biosynthesis of flavones and flavanols, *CmPAL* and *CmCHS*, were upregulated, while genes linked to the biosynthesis of anthocyanidins, *CmDFR* and *CmANS*, were downregulated (Figure 7B). Additionally, genes linked to the biosynthesis of lycopene, *CmPSY* and *CmPDS*,

were markedly downregulated, while genes linked to the biosynthesis of lutein, *CmLCYE* and *CmECH*, were upregulated, and *CmCCD4*, which represses carotenoid biosynthesis, was downregulated (Figure 7B). These gene expression changes resulted in the loss of red coloration and favored the presentation of bright yellow coloration in YM flowers.

## 4. Discussion

Although bud sport mutations are relatively rare, they often result in distinctive, and potentially valuable, qualities that differ from the parent plant. Because of this, bud sports are considered a useful germplasm resource for the development of novel varieties of plants. Chrysanthemums have been cultivated for more than 2000 years; indeed, countless varieties of these flowers have been produced through bud sport mutations [24]. In this study, we identified a novel, yellow-flowered bud sport mutation (YM) from a normally pink-flowered chrysanthemum variety ('Pink Candy', PC). Because this variety has been selected by China Agricultural University for its edible and medicinal qualities, we sought to examine the biochemical differences between PC and YM flowers, hoping to identify and characterize the novel bud sport and provide a foundation for the production of a novel chrysanthemum cultivar.

First, the content of several primary substances, including soluble sugars, soluble proteins, and polyphenols, was higher in YM than PC flowers (Figure 2B–D). As soluble sugars and proteins are well known for their role as essential nutrients [40], and polyphenols are reported to be associated with the classification and identification of taste of tea and other edible plants [41], the differences between the content of these primary substances in two materials suggested potential differences in nutritional ingredients and taste features of PC and YM. Meanwhile, polyphenols can act as strong antioxidants and play a role in the prevention of some degenerative diseases [42,43]; Flavonoids, a specific class of polyphenols including naringenin, galuteolin, and baicalin, not only act as pigments but are also involved with numerous physiological processes, including pathogen resistance [44,45]. Therefore, the different content of these metabolites (Figure 3) indicated the potential differences in medicinal value of the two flower types. Furthermore, some of specific aromatic substances of inflorescence of PC and YM varied greatly (Figure 4), indicating the potentially different fragrances of PC and YM flowers. Taken together, we conclude that the bud variant YM is likely to have different edible and medicinal effects from the parent plant PC. Nevertheless, the invisible changes affecting the edible and medicinal qualities of YM require many more sensory tests and medicinal evaluation in the near future.

The most obvious difference between PC and YM was their flowers color, which was another interesting aspect. Our analysis of the pigment compounds of the two flower types revealed results consistent with their visible color differences;, for example, PC flowers contained a greater proportion of anthocyanins and YM flowers contained a greater proportion of carotenoids and lutein (Figure 5). Similarly, the relative expression levels of key genes linked to flavonoids biosynthesis, such as *CmPAL*, *CmF3H* and *CmDFR*, as well as the carotenoid biosynthesis genes, such as *CmLCYE*, *CmECH*, *CmPSY*, *CmPDS*, *CmNXS*, and *CmCCD4*, were significantly different in PC and YM flowers (Figure 6A,B), which was in alignment with pigmentation analysis. Notably, the carotenoid cleavage dioxygenase 4 (*CCD4*) gene, which was suppressed by 83.22% in YM flowers, is crucial for regulating carotenoid degradation [46]. Specifically, the expression of CCD4 leads to a significant decrease in carotenoid biosynthesis in plants [29]. In white chrysanthemums, the suppression of *CmCCD4a* leads to yellow flowers [38,47]. Similarly, in a pink-to-orange flower color mutant chrysanthemum, the lower expression of *CmCCD4a-5* leads to the accumulation of carotenoids in orange ray florets [28]. We suggest that the potential driver of the yellow coloration of YM flowers is the very low expression level of *CmCCD4*.

In this study, we identified and characterized a novel color changed bud sport. In addition to the obviously changed petal color, some of inner substances and fragrance of bud sport were also altered. However, the causes and underlying molecular mechanism resulting in the novel bud sport remain a further focus of our research.

## 5. Conclusions

Overall, we found several notable biochemical and gene expression differences between YM and PC flowers. The profiles of both primary and secondary metabolites, including soluble sugars and amino acids, polyphenols, apigenin, naringenin, and galuteolin, were significantly altered in YM flowers, suggesting potential differences in stress resistance between the two materials. The content of several aromatic compounds was also significantly altered in YM flowers, indicating differences in the scent profile between the two flower types. Differences in the proportion of flavonoids, anthocyanins, and carotenoids between PC and YM flowers resulted in their striking difference in color. Furthermore, the altered expression levels of genes such as *CmPAL*, *CmCHS*, *CmF3H*, *CmFLS*, *CmLCYE*, *CmECH*, and *CmCCD4* were responsible for alterations in the floral content of these pigment compounds. These results provide a foundation for the production of a novel chrysanthemum cultivar.

**Supplementary Materials:** The following are available online at https://www.mdpi.com/article/10.3390/agriculture12091323/s1, Figure S1: The color reaction of flavonoids in the ray florets of PC and YM. The experiment of petroleum ether (A), HCl (B), $NH_3 \cdot H_2O$ (C), $Al_2O_3$(D), $(CH_3COO)_2Pb \cdot 3H_2O$ (E), $FeCl_3$(F), $H_2SO_4$ (G), $NaBH_4$ (H), $Na_2CO_3$ (I), $H_3BO_3$ (J), $SrCl_2 \cdot 6H_2O$ (K) in the ray florets of PC and YM; Figure S2: The HPLC chromatograms of the standards in the ray florets of PC and YM(A).The relative abundance of chlorogenic acid (B), isochlorogenic acid (C), acacetin (D), apigenin (E), quercetin (F), rutin (G), hesperetin (H), naringenin (I), galuteolin (J), luteolin (K) and baicalin (L) in ray florets of PC and YM; Table S1: Primers used in this study.

**Author Contributions:** Conceptualization, B.H. and C.-H.S.; methodology, L.-D.D. and Y.-H.L.; formal analysis, Y.-H.L.; investigation, L.-D.D., J.-Z.L., X.-Q.D.; data curation, L.-D.D., X.-Q.D.; writing—original draft preparation, C.-H.S. and L.-D.D.; writing—review and editing, C.-H.S., D.-G.H. and L.-D.D.; supervision, C.-H.S. and D.-G.H.; project administration, C.-H.S.; funding acquisition, C.-H.S. All authors have read and agreed to the published version of the manuscript.

**Funding:** This work was supported by grants from National Natural Science Foundation of China (31902049) and the National Key Research and Development Program (2018YFD1000405).

**Institutional Review Board Statement:** No applicable.

**Informed Consent Statement:** No applicable.

**Data Availability Statement:** All data generated or analyzed during this study are included in this published article.

**Acknowledgments:** We would like to thank Bo Hong at China Agricultural University for kindly providing the chrysanthemum cultivar ('Pink Candy') and Da-Gang Hu at Shandong Agricultural University for his critical reading of the article during the preparation of this manuscript. We also would like to thank Fangfang Ma and Zhilong Bao at Shandong Agricultural University for experimental platform assistance.

**Conflicts of Interest:** The authors declare no conflict of interest.

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
