# Peer review of "Identification and Characterization of Petal Color Change from Pink to Yellow in Chrysanthemum morifolium ‘Pink Candy’ and Its Bud Variant"

_agriculture, doi:10.3390/agriculture12091323_

Round 1

Reviewer 1 Report

The manuscript number: agriculture-1877718 entitled “ Identification and characterization of petal color change from pink to yellow in Chrysanthemum morifolium ‘Pink Candy’ and its bud variant” authored by  Du  et al., is a comparative study of color development in chrysanthemum.  The authors analyzed the biochemical and molecular to evaluate the color development. The reviewer gone through the manuscript and found that the manuscript is presented as per journal guidelines, the contents of manuscript is in define format, table and figures are presented appropriately. Followings are the query and suggestion for improving the quality of the manuscript-

Abstract

1.     Abstract is more introductory, emphasize on results and there are many grammatical mistakes such as use of “as” (as one of the most popular ornamental), “on” (The difference on the appearance), “of” (the difference of primary substances) “were” (parent plant were much more significant) etc..

2.     Write full name of CmPAL, CmDFR, CmF3H, CmNXS and CmCCD4 genes

Keywords

1.     Add few catchy words like pigment biosynthesis, gene expression

Introduction

1.     Rewrite “Flower ornamental qualities comprise …….trait in ornamental” for better clarity

2.     Delete “ Over the years, a ……..and regulation of flower color”

3.     Rewrite “Chrysanthemum (Chrysanthemum morifolium), as one of the most economically valu-able ornamental plants ….. varieties and abundant variations.” For better clarity.

4.     It is better to add a global scenario of  Chrysanthemum cultivation and economic value.

5.     Rewrite “Here, we reported the discovery of a …..yellow, rather than pink, ray florets.” For better clarity

Method and Materials

1.     Replace “introduced” with “obtained or procured”

2.     Replace “under same cultivation” with “under same condition”

3.     It is better to use “estimated” in place of “measured”

4.     Some section of biochemical analysis can be merged to reduce the number of subheading

5.     Overall the method and material section written nicely but there is use of many unscientific words.

Results

1.     Some subheadings of results also may be merged to reduce the subheading numbers

2.     Delete “the normally”

3.     Delete “process”

4.     Replace “adaptive differences” with “adaptive changes”

5.     Replace “aroma compounds” with “aromatic compounds” throughout the manuscript

6.     Rewrite “The color reaction suggested that pigments such as flavonoids, anthocyanins, carot-enoids and lutenin were the potential cause determining the color difference between ……… respectively (Figure 5A and 5B).” for better clarity.

7.     Replace “Petal pigmentation is closely related” with “Petal pigmentation is directly related”

8.     Replace “To ascertain the molecular” with “To understand the molecular”

9.     What is mean by “CmPAL, was significantly unregulated in YM flowers”

10.  The results are presented and written in a nice way

Discussion

1.     Replace “for the production of novel varieties” with “for the development of novel varieties”

2.     The author should interpret their biochemical findings and their role in color development and also should focus on color development pathways and role of gene expression patterns in color development.

3.     The authors presented lot of data in support of their findings but the discussion section is poorly written and the interpretation of results in discussion is also very poor. Therefore, it should be rewritten for the manuscript quality improvement.

Conclusion

1.     Conclusion is ok but it can be improved for better soundness.

Figures

2.     Figure 1-7 are ok but Figure 7B should be improved for better clarity.

3.     Supplementary figure  1 and 2 are also ok

Tables

1.     Table 1 is ok

2.     Supplementary table S1 can be shifted to the manuscript

Overall the present manuscript prepared and presented very nicely, the authors presented huge information in support of their findings. There are very few reports available about pink to yellow color change in chrysanthemum. The manuscript may be accepted for the publication in “agriculture” only after addressing the queries raised or modification suggested. There is need of minor revision for manuscript quality and impact improvement.

Thanks.

Author Response

Thanks for your suggestions.

We modified our manuscript according to your suggestions. Please check the attached file and reffer to the new manuscript.

Reviewer 2 Report

Although the work is novel and the experimental design is precise, the manuscript's writing is lacking. The references 32, 47, and 48 from the literature are absent from this document. Additionally, the manuscript's discussion part is seriously deficient. Authors should properly communicate their findings. One of my key questions is: Why, after conducting research on ray florets, is the author discussing stress tolerance here? Is there a role for ray florets (different colours and other metabolites) in abiotic and biotic stress tolerance or resistance? The author should make it clear that they did not test these plants (wild type and mutant) under any stress. Although flavonoids and anthocyanins are the main metabolites that contribute to stress resistance/tolerance against stresses in plants. but in this study, these metabolites are significantly higher in PC than in YM. So how can you claim that YM is more tolerant/resistant to stress?

Author Response

Thank you for your suggestions and the good question. We answered your question and modified our manuscript according to your suggestions. The reference 32 , and references 47, 48 were corrected in the new manuscipt.

Please refer to the new manuscript and the attached coverletter.

We are sorry for the misunderstand on this question. We did not conduct any researches on stress tolerance or resistance. Based on previously reported studies, we discussed the possibility of stress tolerance of YM. We did not claim that possibility for certain. Of course, as we said in manuscript, this theory required further verification”.

Honestly, we did acknowledge the increased metabolites in ray florets may be not enough to contribute to stress resistance of whole plants. Therefore, we rewrote the discussion part for better clarity.

Thanks again on this question.

Round 2

Reviewer 2 Report

The authors have incorporated almost all queries into this manuscript, hence it is accepted for publication.